# PERL: a dataset of geotechnical, geophysical, and hydrogeological parameters for earthquake-induced hazards assessment in Terre del Reno (Emilia Romagna, Italy)

Chiara Varone[1], Gianluca Carbone[1], Anna Baris[2], Maria Chiara Caciolli[1,3], Stefania Fabozzi[4], Carolina Fortunato[1], Iolanda Gaudiosi[1], Silvia Giallini [1], Marco Mancini[1], Luca Paolella[2], Maurizio Simionato[1], Pietro Sirianni[1], Rose Line Spacagna[1], Francesco Stigliano[1], Daniel Tentori[1], Luca Martelli[5], Giuseppe Modoni [2], Massimiliano Moscatelli [1]

[1] CNR Italian National Research Council, Institute of Environmental Geology and Geoengineering (IGAG), RM1, 00015 Montelibretti, Italy
[2] University of Cassino and Southern Lazio, Dept. of Civil and Mechanical Engineering, Cassino 03043, Italy
[3] University of Perugia, Department of Physics and Geology, Perugia 06100, Italy
[4] Ministero delle Infrastrutture e dei Trasporti, Roma 00157, Italy
[5] Geological, Seismic and Soil Survey, Emilia-Romagna Region, Bologna, 40127, Italy

*Correspondence to*: Chiara Varone (chiara.varone@igag.cnr.it; chiara.varone@cnr.it)

**Abstract.** In 2012, the Emilia Romagna Region (Italy) was struck by a seismic crisis characterized by two main shocks (ML 5.9 and 5.8) which triggered relevant liquefaction events. Terre del Reno is one of the municipalities that experienced the most extensive liquefaction effects due to its complex geo-stratigraphic and geo-morphological setting. This area is indeed located in a floodplain characterized by lenticular fluvial channel bodies associated to crevasse and levee clay–sand alternations, related to the paleo-Reno River. Therefore, it was chosen as case study for the PERL project, which aims to define a new integrated methodology to assess the liquefaction susceptibility in complex stratigraphic conditions through a multi-level approach. To this aim, about 1800 geotechnical, geophysical and hydrogeological investigations from previous studies and new realization surveys were collected and stored in the PERL dataset. This dataset is here publicly disclosed and some possible applications are reported to highlight its potential.

## 1 Introduction

In these last years, an increasing number of source data are publicly disclosed, allowing a wider access to research activities. Key examples are the huge amount of free satellite imagery (i.e., Sentinel, Landsat) provided by the main space agencies and the cutting-edge tools and procedures integrated in widely known and open source EO platforms such as Google Engine. A multitude of algorithms and codes are available for all the fields of knowledge concerning natural hazards, while their application is made easier by the increasing number of open-access inventories of natural phenomena (i.e., Martino et al. 2014; Guarino et al., 2018; Tanyaş et al., 2022). However, only a few examples of datasets of in-situ investigations and related

parameters are publicly disclosed, and this should be a gap to be filled. With regard to macro-types of investigations (i.e., geological, geophysical, geotechnical, hydrogeological, etc.), some databases are currently available worldwide (i.e., Orgiazzi et al., 2017, Kmoch et al., 2021, Geyin et al., 2021, Minarelli et al., 2022), as well as for the Italian national territory. An

example is provided by Vannocci et al. 2022, which includes geotechnical and hydrological soil parameters for shallow landslide modelling.

However, there are only a few examples of freely available products which integrate different macro-typologies of in-situ investigations in a unique database, especially with reference to the Italian territory (Gaudiosi et al., 2021). In the light of the above, the aim of the authors is to make freely available a dataset of about 1800 geological, geophysical, geotechnical and

hydrogeological in-situ investigations and related parameters collected in the Terre del Reno municipality (Emilia – Romagna Region, Italy). The study area is affected by severe seismic hazards and prone to seismically induced effects, as extensively documented by the 2012 seismic sequence which was characterized by more than 2000 earthquakes (Facciorusso et al., 2016). Two main shocks were recorded during the crisis: the first one on 20th May with ML 5.9 and epicenter in Finale Emilia, and the second one on 29th May with ML 5.8 and epicenter in Medolla, both in Modena Province.

As widely reported in bibliography, the propagation of seismic waves through the upper portion of the soil can be modified by local site conditions (i.e., Bozzano et al., 2017; Fabozzi et al., 2021; Falcone et al., 2020,2021; Gautam 2017; Luo et al., 2020; Meza-Fajardo et al., 2019) and can determine the triggering of earthquake-induced effect at ground surface (i.e., Forte et al., 2021; Martino et al., 2017, 2019; Giannini et al., 2022; Paolella et al., 2022; ). In Terre del Reno, these earthquakes triggered several earthquake-induced effects (Chini et al., 2015, Papathanassiou et al., 2015), among which linear and punctual

liquefaction effects were the most prominent. These effects may occur when saturated granular deposits are shaken by a seismic action, and their magnitude depends on the combination of earthquake intensity and soil condition. Literature reports plenty of liquefactions events happened in complex geological conditions and triggered by earthquakes with various magnitudes such as for instance Gorkha, Nepal (Gautam et a., 2017), Christchurch, New Zealand (Maurer et al., 2019), Urayasu, Japan (Baris et al., 2021), 2008 Wenchuan, China (Zhou et al., 2022) and 2019 Dürres, Albania (Mavroulis et al. 2021) earthquakes.

Liquefaction effects in Terre del Reno were mainly related to the complex sedimentological and stratigraphic setting of the areas (i.e., Stefani et al., 2018, Tentori et al., 2022), characterized by multiple and alternate sandy and silty-sandy packing hosting local (shallow) and regional (deep) aquifers (Regione Emilia – Romagna, 1998). Several authors (i.e. Ecemis 2021; Jain 2022) highlighted that the presence of a tiny alternation of silt and sands seems to influence the liquefaction occurrence while other studies focused on the role of silty sands and soil packing condition on liquefaction triggering (i.e., Naeini et al.,

2004, Stamatopoulos et al., 2010, Gobbi et al., 2022a). To overcome the difficulties related to heterogeneously complex soil conditions, integrated approaches are applied to predict the occurrence of liquefaction by combination of numerical and experimental methods (Gobbi et al., 2022b; Rios et al., 2022; Paolella et al. 2022). Further steps toward this direction were made for the Terre del Reno case study by pursuing two main objectives of PERL project in order to: i) define a new integrated methodology to assess the liquefaction susceptibility in complex stratigraphic settings through a multi-level approach; ii)

perform the seismic microzonation of the municipality for land and civil protection planning purposes. This project allowed

the collection and analysis of the above mentioned in-situ investigations, and the elaboration of thousands related parameters that were stored in a harmonized and standardized dataset (named "PERL") conceived to guarantee interoperability with existing ICT solutions and data models. The availability of such a dataset of surveys, catalogued and processed according to shared standards, makes Terre del Reno one of the best-characterized municipalities in Italy in terms of seismic hazard and

earthquake-induced effects. This flexible dataset can be manipulated and combined to tackle different problems and represents a powerful resource for the scientific community, for those who cannot set-up and manage a living laboratory or directly perform on-site investigations.

For these reasons, authors provide complete access to the dataset through the supplementary materials and present two different applications herein used as references to highlight the potential of PERL dataset.

**2 Geological setting**

*Structural and Stratigraphic Setting*

The study area is located within the southern portion of the Po alluvial plain, which represents the sedimentary cover of the Po Basin infill (Fig. 1). The geological substrate of the study area, which lies along the northern sectors of the Apennine chain, shows complex fold and thrust structures with arcuate geometry associated with strongly asymmetrical foredeep basins.

Although the Po Basin represents both the Alpine retroforeland basin and the Apennine foredeep, its Cenozoic structural evolution was mainly driven by the north-east migration of the external front of northern Apennines, which consists of four arcuate fold-and-thrust systems: the Monferrato Arc, the Emilia Arc, the Ferrara Arc and the Adriatic Arc (Pieri and Groppi, 1981; Royden et al., 1987; Scrocca et al., 2007). These systems that are buried beneath the present Po plain, were active since the Late Miocene (Fig. 1) and are still considered seismogenic (Boccaletti et al., 2011; Ghielmi et al., 2013). In particular, the

movement of a segment of the Ferrara Arc thrust system (i.e., the Mirandola thrust system) was responsible for the 2012 Emilia seismic events (IsideWorking Group, 2010), which triggered numerous co-seismic effects associated with liquefaction phenomena in the Provinces of Ferrara, Modena, and Bologna. In the study area, the shallowest Quaternary sedimentary fill consist of marine deposits (Marine Quaternary in figure 1b) and 100 ky-spaced transgressive–regressive cycles constituted by nearshore sands and alluvial deposits, formed during interglacial and glacial periods respectively (Continental Quaternary in

figure 1b). The stratigraphic framework of the topmost late Pleistocene to Holocene Po Basin succession (at 0-40 m depth from the ground surface), documents a succession of tabular-shaped fluvial sands (i.e. glacial) overlain up-section by the Holocene poorly drained- and mud-rich floodplain and swamp/marsh succession with subordinate lenticular fluvial sandy channel bodies associated to crevasse and levee clay–sand alternations fed by the paleo-Reno River (i.e. interglacial) (Bruno et al., 2021; Stefani et al., 2018; Tentori et al., 2022). The Reno River modern drainage basin extends for about 2500 km$^2$ in

the Northern Apennines. Owing to the low topographic gradients in the area, the paleo-Reno River experienced fast aggradation and frequent avulsion episodes during recent and historical times (see Tentori et al. 2022 and references therein).

*Hydro-stratigraphic Setting*

The hydro-stratigraphic architecture reflects the depositional and tectonic evolution of the southern Po sedimentary basin from Pleistocene to Holocene (Molinari et al., 2007; Emilia-Romagna Region and ENI-AGIP, 1998). The aquifers from the most superficial hydro-stratigraphic group (e.g., Group A), consist of six lower-order hydro-stratigraphic units belonging to the Quaternary fluvio-deltaic and alluvial depositional systems. In the study area, Group A aquifers consist of the sandy fluvial bodies deposited during glacial periods, separated by the muddy-dominated intervals of transgressive alluvial facies (aquitards) deposited during interglacial periods. The more surficial composite aquifer system named A0 by Molinari et al. (2007) consists of two sandy-dominated aquifer units hosted within the late Pleistocene-Holocene channelized bodies and encased by alluvial floodplain muds. Based on the piezometric level dating back to summer 2012, Calabrese et al. (2012) placed the groundwater level of the shallower semi-confined aquifer at about 3-4 m depth below the levee and about 1-2 m in the floodplain.

## 3 Materials and methods

The PERL dataset was obtained by merging three databases provided by different institutions. Additional 17 geotechnical investigations were specifically performed in the framework of the PERL project.

The three existing databases are:

- MUDE database (Modello Unico Digitale per l'Edilizia – Unique Digital Model for Building)

The MUDE database consists of 384 records including punctual and linear in-situ investigations. Data were extracted from a series of technical reports produced to plan the reconstruction works of buildings collapsed during the 2012 seismic crisis. Since the digital formats of these investigations were originally not available, geo-localization, key information and measured parameters were obtained from the digital scans of technical and geological reports.

- RER database (Regione Emilia-Romagna - Emilia-Romagna Region database)

The RER database is composed of 906 geo-localized, punctual records, associated with a set of keys information (typology, date, coordinates, and maximum depth) and a scan of the investigation sheet. Parameter were extracted from investigations sheets as they are not available in a digital format. This database is available at https://servizimoka.regione.emilia-romagna.it/mokaApp/apps/geg/index.html (last access: 12/09/2022).

- SM database (Seismic Microzonation Studies)

The SM database is composed of 1284 records including punctual as well as linear in-situ investigations. These investigations are geo-localized and organized in a standardized structure according to *Commissione tecnica per la microzonazione sismica* (2015). The key information (typology, date, coordinates, etc.) of each investigation are stored in a dedicated table, while all the measured parameters are reported in chained tables. This database is available at https://www.webms.it/ (last access: 12/09/2022).

The first problem faced when merging these databases was the presence of duplicate information. To avoid duplicates, a methodology to discern and verify the uniqueness of an investigation was elaborated.

This methodology is based on the implementation of a series of multiple, progressive True/ False (TF) controls applied to various control parameters (CP) relative to all the investigations included in the pertinence area. The latter was defined as a circle with a radius equal to 200 m centered in correspondence of the considered investigation. The progressively considered CP (Fig. 1) are: CP1) absence of another investigation within the area of pertinence; CP2) un-matching of the investigation typology; CP3) un-matching of the date of survey; CP4) matching of maximum depth reached by the investigation. Each CPm (m=1,2,3,4) is checked in a dedicated TF test (TFn with n=1,2,3,4). Starting from TF1, an investigation that verifies CP1 is moved to TF2 for CP2 verification up to TF4. Each time that a CPm in a TFn is not verified, the investigation is defined as "unique". If an investigation verifies all the control parameters, it is defined as "redundant" and removed from the database. The application of this methodology allowed to identify and remove 32% of the investigations, obtaining a final dataset composed of 1805 unique investigations (Fig. 2).

## 4 Data description

The PERL dataset consists of two shapefiles implemented into a GIS system and an associated geodatabase. The two shapefiles are named *ind_pc* and *ind_ln*, and correspond to punctual and linear investigations, respectively (EPSG:32633). The associated attribute tables contain the main information of each investigation:

- *ID:* unique identification number for each investigation;
- *Investigat:* investigation typology.

The complete set of investigations and the related measured parameters are reported in an Excel file following this structure:

- *ID:* unique identification number of each investigation
- *Type_par:* parameter typology (see 'list of parameters' for legend)
- *Value:* parameter value
- *Depth_top:* depth (m) of the layer top to which the parameter refers.
- *Depth_bottom:* depth (m) of the layer bottom to which the parameter refers.

The PERL dataset includes 1805 records corresponding to as many investigations, and guarantees an average density of 35 punctual (Fig. 3a) and/or linear measurements (Fig. 3b, c) per square kilometer of Terre del Reno municipality (about 51 km$^2$). Focusing on investigations typologies (see the list of typologies and codes and Fig. 4a), the dataset consists of 71% of penetrometer tests (CPT, CPTU, CPTE, SCPT, SPT, DN), 16% of boreholes and trenches (S, T, SP, SC, SD), 12% of punctual and linear geophysical investigations (CH, DH, HVSR, MASW, ESAC_SPAC), 1% of laboratory and hydrogeological tests (CR, CI, CD, SM, ED, TD, LF) (Fig. 3).

Penetrometer tests, geognostic boreholes, trenches, and borehole geophysical tests are characterized by a depth of investigation ranging from few meters to more than 100 m (maximum depth: 265m) (Fig. 4b). About the 90% of them, reach a maximum depth of investigation of 35m. Thus, the most represented depth classes are 30-35m and 10-15m with 330 (21%) and 310

(20%) investigations, respectively. Penetrometer tests are characterized by depths ranging between 5 and 50m, with the 30-35m class being the most represented. On the contrary, boreholes and trenches cover the entire spectrum of the dataset depth. However, it is worth noticing that about 60 boreholes and trenches reach a depth higher than 55m, which is the most represented classes together with the 10-15m class. Penetrometer tests, boreholes, trenches, and geophysical tests are characterized by investigation depths ranging from few to some hundred meters with a maximum of 265m. About the 90% of them, reach a maximum depth of investigation equal to 35m. Most investigation are carried out up to a depth of 30-35m and 10-15m (330 (21%) and 310 (20%) investigations for each class respectively). On the contrary, boreholes and trenches cover the whole spectrum of depth classes. However, it is worth noticing that about 60 boreholes and trenches reach a depth higher than 55m, which is the most represented class together with the 10-15m class.

## 5 Examples of applications

To address some of the conceptual points discussed in the Introduction and better highlight the uniqueness and potential of the PERL dataset, we present two different applications. In the first case study, we take advantage of the PERL database to represent the complex geology beneath the San Carlo alluvial plain. The second case history focusses on a statistical inference of the PERL geophysical data to obtain soil dynamics when experimental information are missing.

### 5.1 Stratigraphic reconstruction of liquefiable layers thickness in the San Carlo subsoil

The PERL database includes several sedimentological, geotechnical, geophysical, and hydrogeological data which can be used to reconstruct the stratigraphic architecture of the Terre del Reno subsurface and provide a reliable geological framework for future studies devoted to earthquake-induced hazards mitigation. The position and the thickness of liquefiable portion within the subsoil are key information for liquefaction risk assessment and mitigation. The possibility of automatically build three-dimensional subsoil model with advanced procedures represents a current topic of the applied technological research. Here, a combination of these two approaches is presented to spotlight the potential of PERL dataset.

As an example, Figure 5 shows the geostatistical interpolation, performed with the Ordinary Kriging, of the Cumulated Thickness of the Liquefiable layers (CTL) in the district of San Carlo. In particular, the CTL has been manually extracted from 33 boreholes and automatically obtained on 148 CPTs by applying the procedure proposed by Spacagna et al. (2022). The obtained map has been overlayed on the map of liquefaction evidence that occurred after the Emilia-Romagna 2012 seismic sequence, showing a good match between the liquefaction-induced surficial manifestations and the CTL distribution.

## 5.2 Statistical analysis of shear waves variability with depth

As widely represented by literature data, the amount of available investigations progressively decreases with soil depth. Thus the uncertainty in subsoil characterization increases from the ground surface to the deepest layers of the soil. To overcome this issue, an example of statistical inference of soil parameters is presented. Based on the available data, a correlation analysis between the values of $V_s$ (m/s) and depth (m) was carried out to infer information when the depth of investigation does not guarantee a correct soil parameterization. Considering the high geological and stratigraphic complexity of the case study, a lithological-based statistical inference was performed. Specifically, for each lithology L, the scatter plot of the value of $V_s$ (m/s) as a function of depth, D (m) and the corresponding linear regression was calculated.

The linear regression model is defined by the following relationship:

$$V_S(D) = aD + V_{S0}$$

where $a$ and $V_{s0}$ correspond to the slope and intercept of the model line, respectively.

PERL database can rely on 164 $V_s$ profiles mainly identified from penetrometer tests (SCPT), Down Hole (DH), MASW, ESAC_SPAC, SDMT, and Cross Hole (CH) tests. Each of these $V_s$ profiles was discretized with a step size of 1 m in depth and, through an automated procedure, each meter of depth was associated with a lithological (L) information extracted from proximal boreholes.

For each lithology L, a statistical analysis was performed, and the linear regression models were calculated. The results obtained for MH and SP lithologies are here presented as case examples (Fig. 6). The corresponding values of coefficients $a$ and $b$, and the coefficient of determination $R^2$ are reported in Table 1.

| Lithology (L) | $a$ | $V_{s0}$ | $R^2$ |
|---|---|---|---|
| MH | 4.18 | 132 | 0.59 |
| SP | 1.28 | 195 | 0.79 |

**Table 1: Coefficients (a= slope, $V_{s0}$= intercept) of the regression model for 2 lithologies (L) and related coefficient of determination ($R^2$)**

As expected, the obtained $a$ and $b$ values highlight a positive slope as the depth mean value increases together with the mean of the dependent variable $V_s$. At the same time, these values allow to quantify how much the $V_s$ value changes per meter of depth showing an SP variation rate greater than that characterizing MH. Moreover, the coefficient of determination $R^2$ obtained for MH and SP lithologies are characterized by a 0.59 and 0.79 value, respectively, proving the reliability of the fitting. When experimental data are lacking, for depths that fall within the variability range of the available data, the regression models allow to obtain $V_s$ by interpolation while an extrapolation can be applied for grater depths.

215 Results may be used in the future for comparison with other Italian estimates (Romagnoli et al., 2022) or, combined with ambient vibration measurements to define the thickness of the resonant sedimentary layers (D'Amico et al., 2008, Giannini et al., 2021).

## 6 Conclusions

As part of PERL project, a considerable number of investigations were collected in the Terre del Reno municipality (Emilia 220 Romagna Region (Italy). This area experienced the most extensive liquefaction effects during the 2012 Emilia Romagna seismic crisis and remains exposed to severe seismic hazards and seismically induced effects due to its complex geological setting.

Thanks to this study, a complete and free access to the PERL dataset, which includes 1805 punctual and linear *in-situ* investigations consisting of geological, geotechnical, geophysical and hydrogeological data is provided. The database is 225 composed of 71% of penetrometer tests, 16% of boreholes and trenches, 12% of geophysical investigations and 1% of laboratory and hydrogeological tests.

Two applications of the PERL dataset are presented to highlight its potentiality and to show that high-quality large dataset could be critical to infer information in areas and/or portions of the soil characterized by poor or sparse data. The first examples pointed out the database potentials in overcoming problems due to the uneven distribution of surveys across the territory, while 230 the second one spotlight its capability to provide information at subsoil depth not reached by investigations.

Other major outcomes of the PERL project in the Terre del Reno municipality included: i) a detailed reconstruction of the subsoil geology to examine the stratigraphic control on earthquake-induced liquefaction (Tentori et a., 2022); ii) data-driven and automatic subsoil characterization through the analysis of CPT-based soil behaviour type (SBT) and soil behaviour type indexes (Ic) combining geostatistical and artificial intelligence genetic approach (Baris et al., 2022); and iii) 3[rd] level seismic 235 microzonation study of the Terre del Reno municipal area (Varone et al., 2022) to mitigate seismic and seismically induced hazards through a sensible urban planning.

Ongoing studies currently focusses on the definition of a comprehensive methodology to quantify liquefaction susceptibility in areas dominated by complex geo-stratigraphic conditions by applying a multi-level approach laying on simplified models and to promote the identification of potentially liquefiable granular bodies, thus mitigating earthquake-induced hazards to 240 allow a sustainable development of this urban area.

**Data availability**

The authors confirm that the data supporting the findings of this study are available within the article and its supplementary materials.

## Code availability

Not applicable

## Author contribution

CV, GC and FS conceptualized the methodology. CV, GC, AB, MCC, SF, CF, IG, SG, LP, MS, PS, RS and DT curated the data. GC implemented the algorithm. CV, AB, DT and RS curated the examples of applications. LM, GM and MM conceptualized and administrated the project. CV wrote the original draft of the paper. CV, GC, AB, MCC, SF, CF, IG, SG, MM, LP, MS, PS, RS, FS, DT, LM, GM and MM reviewed and edited the paper.

## Competing interests

The authors declare that they have no known competing interest.

## 7 Acknowledgements

The authors wish to thank two anonymous referees for their valuable suggestions that improved the quality of the manuscript. This work was carried out in the framework of the project: *"Accordo di collaborazione tra Regione Emilia-Romagna, Amministrazione Comunale di Terre del Reno, Consiglio Nazionale delle Ricerche – Istituto di Geologia Ambientale e Geoingegneria, Università degli Studi di Cassino e del Lazio Meridionale – Dipartimento di Ingegneria Civile e Meccanica, finalizzato a definire una strategia multilivello per valutare il rischio da liquefazione in presenza di argini e situazioni geologiche e morfologiche complesse".* Authors wish to thank Olga Mantovani (Municipality of Terre del Reno) for its support to the project.

## Appendix A : List of investigation typologies and codes

- *CH:* Cross - Hole test
- *CPT:* Cone Penetration Test
- *CPTE:* Electrical Cone Penetration Test
- *CPTU:* Piezocone Penetration Test
- *CR:* Resonant Column Test
- *DH:* Down-Hole test
- *DMT:* Dilatometric Test
- *DN:* Dynamic Cone Penetration Test
- *ESAC_SPAC:* seismic array elaborated by ESAC/SPAC methods

- *HVSR:* ambient noise measurements elaborated by HVSR technique
- *LF:* LeFranc test
- *MASW:* Multichannel Analysis of Surface Waves
- *PA:* borehole (water well)
- *S:* non-destructive borehole
- *SC:* borehole with collection of samples
- *SCPT:* Seismic Cone Penetration Testing
- *SD:* borehole
- *SDMT:* Seismic Dilatometer Marchetti Test
- *SM:* laboratory test on soil sample
- *SP:* piezometer
- *SPT:* Standard Penetration Test
- *T:* trench
- *TD:* shear strength test

## Appendix B : List of parameters and codes

- *AR:* clay classification obtained from laboratory test (%)
- *C:* effective cohesion (MPa)
- *CAM:* number of sample (-)
- *CU:* undrained cohesion (MPa)
- *E1:* index of voids (-)
- *F1:* effective soil friction angle (˚)
- *FR:* resonance frequency (Hz)
- *FS:* skin friction (MPa)
- *G:* shear modulus (MPa)
- *GH:* gravel classification obtained from laboratory test (%)
- *IP:* plasticity index (-)
- *K:* classification obtained from laboratory test (m/s)
- *L:* layer lithology (-)
- *LID:* lithology of the hydrolayer (-)
- *LM:* silt classification obtained from laboratory test (%)
- *PT:* number of SPT blows (-)
- *PTM:* number of DN blows (-)
- *PV: w*eight of the unit of volume (kN/m$^3$)
- *QC:* tip resistance (MPa)
- *SA:* sand classification obtained from laboratory test (%)
- *SG:* water table level (m)
- *U:* hydrostatic pressure (MPa)
- *VP:* compressional waves velocity (m/s)
- *VS:* shear-waves velocity (m/s)
- *W:* water content (%)

**Appendix C : List of lithologies (L) and codes**

- *CH*: inorganic clays with high plasticity
- *CL:* inorganic clays with low- medium plasticity, gravelly or sandy, silty clays
- *GP:* clean gravel poorly graded, mixture of gravel and sand
- *MH:* inorganic silts, fine sands, micaceous or diatomitic silts
- *ML*: inorganic silts, silty or clayey fine sands, silts clayey sands with low plasticity

- *OH:* organic clays of medium-high plasticity, organic silts
- *OL:* organic silt, silty clays with low plasticity
- *PT:* peats and peaty soils
- *RI:* anthropogenic filling
- *SC:* clayey sands, mixture of sand and clay

- *SM:* silty sands, mixture of sand and silt
- *SP:* sands poorly graded
- *SW:* sands well graded, gravelly sands

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

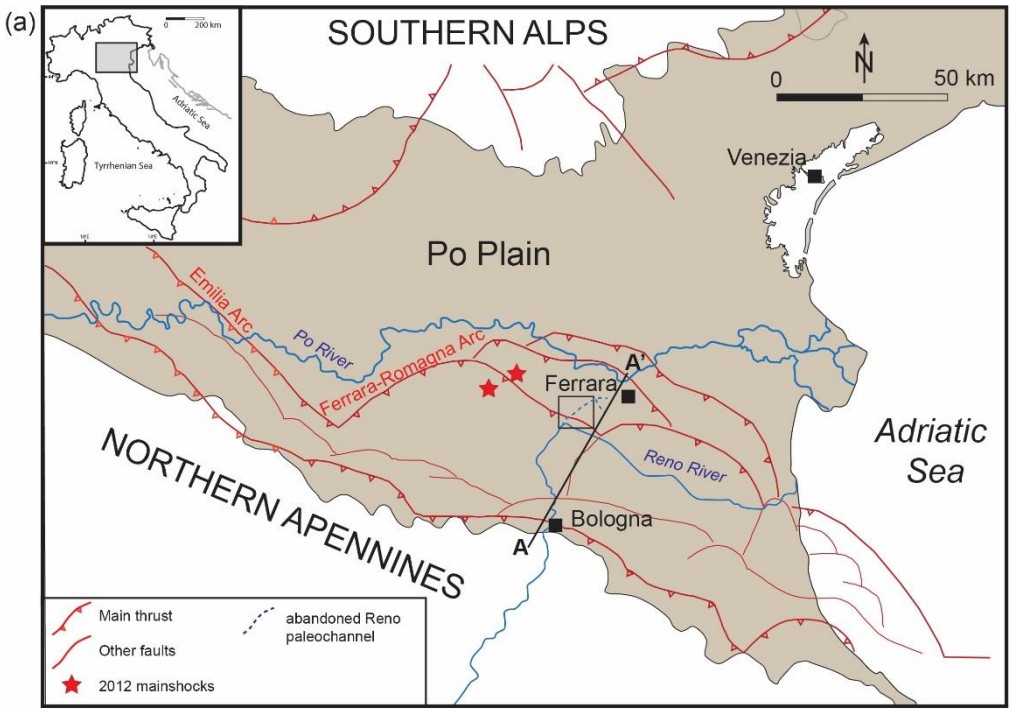

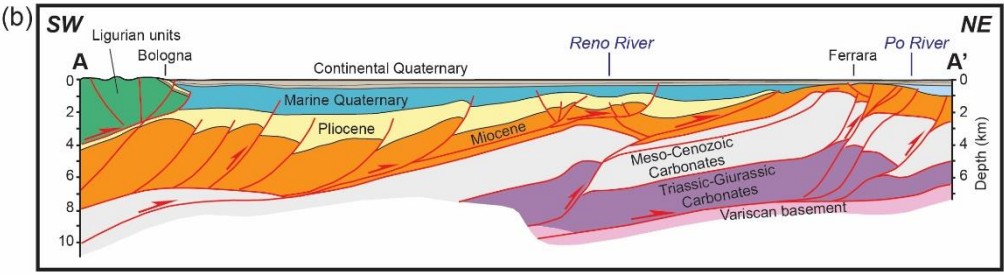

**Figure 1: a) Tectonic sketch map of the Po Plain (Northern Italy) showing the main buried faults of the Northern Apennines and Southern Alps and epicenters of the two mainshock from the 2012 Emilia sequence (red stars) (modified from Tentori et al., 2022 and Bruno et al., 2021). The black-lined rectangle encloses the study area. b) Simplified stratigraphic cross section and major tectonic structures along the trace A–A′ in figure a) (modified after Boccaletti et al., 2004). The tectonic structures are enucleated into Mesozoic to Paleogene carbonate successions, and largely controlled the sedimentary evolution of the terrigenous basins during the Neogene (Ghielmi et al., 2013; Rossi et al., 2015; Ricci Lucchi, 1986). The Pliocene-Pleistocene boundary records the transition from turbiditic sedimentation to marine clay deposition whereas the Quaternary sedimentary fill consists of marine deposits, nearshore sands and alluvial deposits (see text for details).**


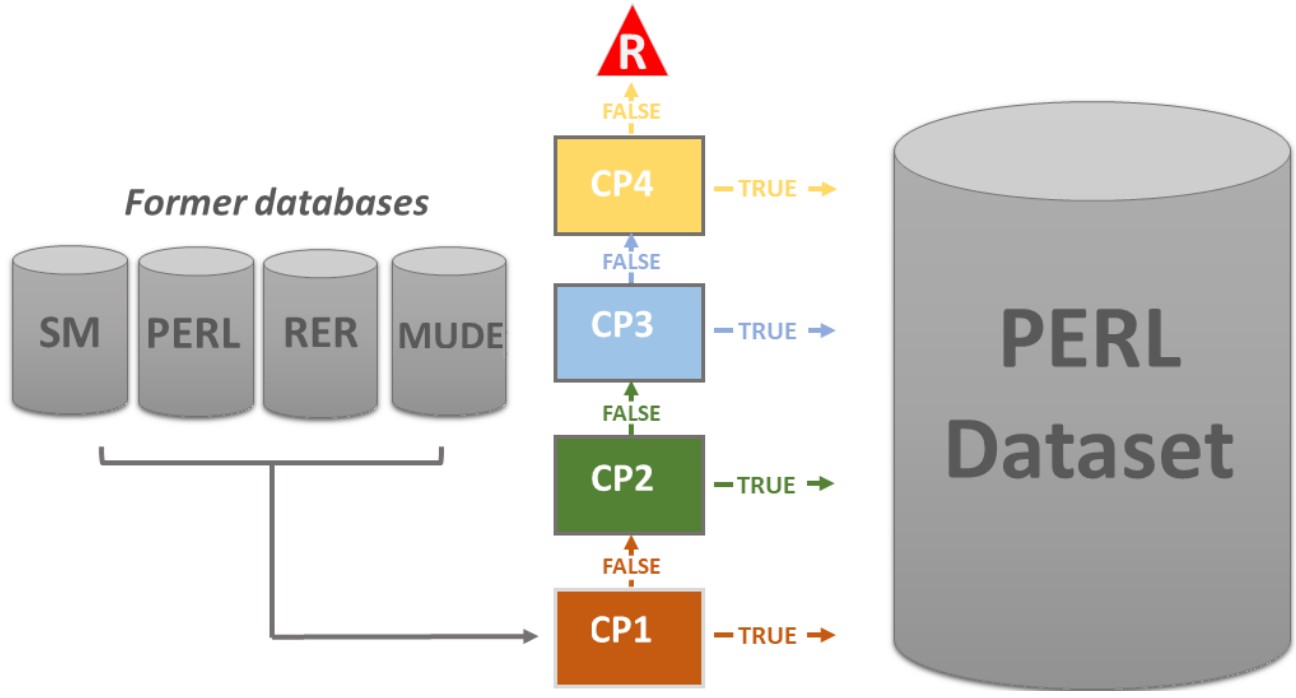

**Figure 2: Synthetic workflow of the method used to merge MUDE, RER, SM database and new realization investigations into the PERL dataset.**




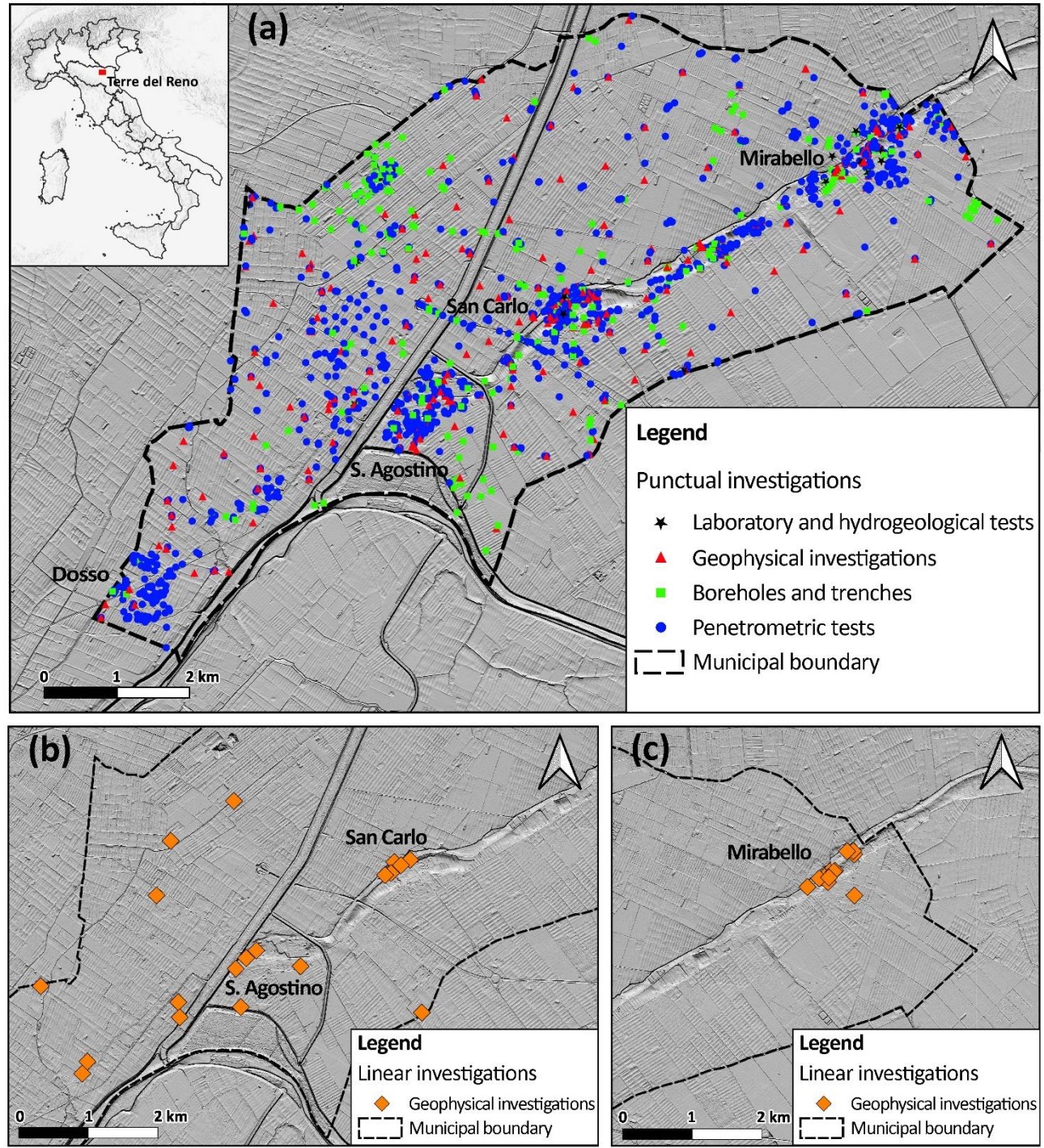

**Figure 3: Spatial distribution of the in-situ punctual (a) and linear (b)(c) investigations composing the PERL dataset. The Digital Elevation Model (DEM) was retrieved from Regione Emilia Romagna (2015)**

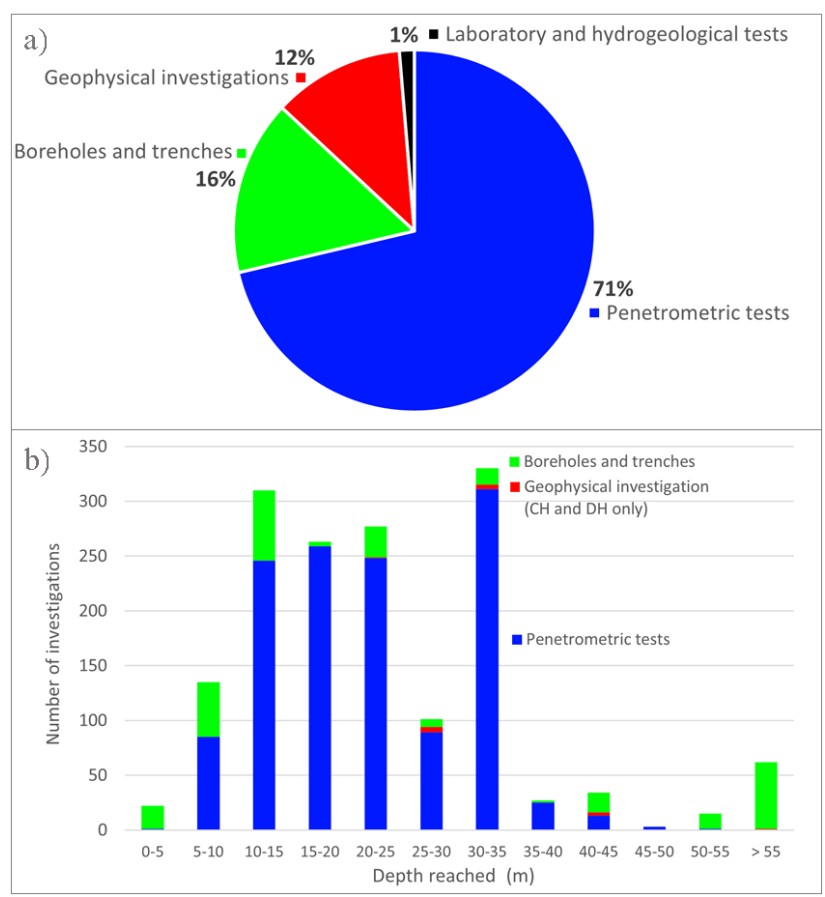

**Figure 4: PERL dataset characteristics: a) classes of in-situ investigations; b) depth reached by penetrometer tests, geophysical investigations (CH and DH), boreholes and trenches.**

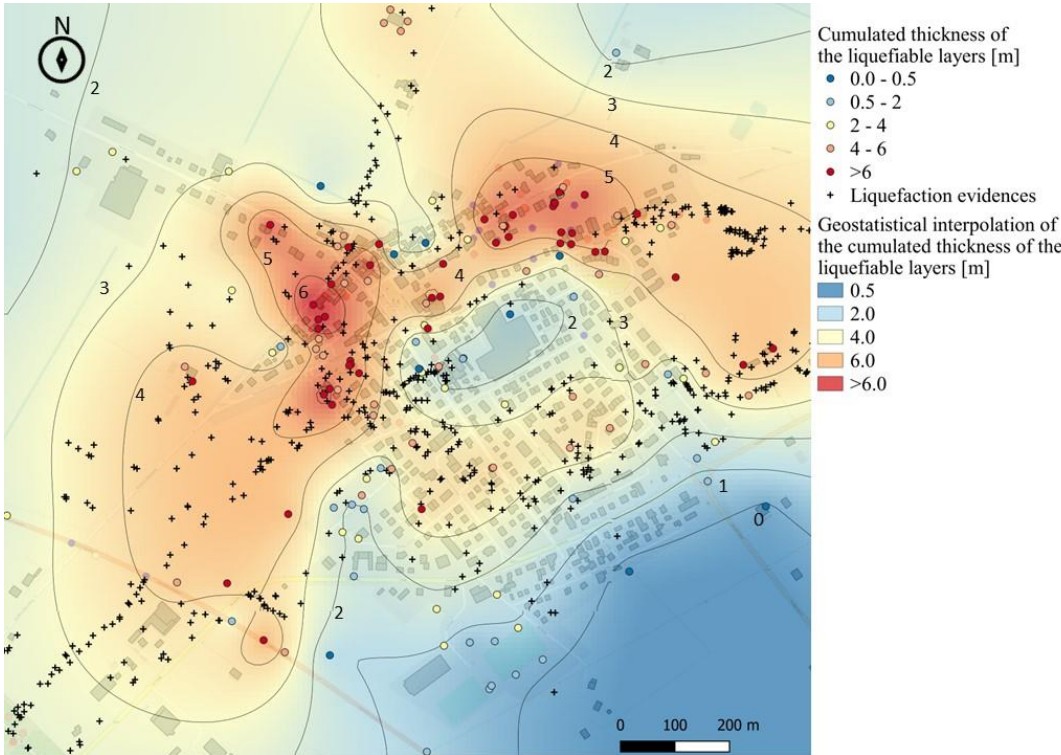

**Figure 5: Geostatistical maps of the cumulated thickness of the liquefiable layer.**


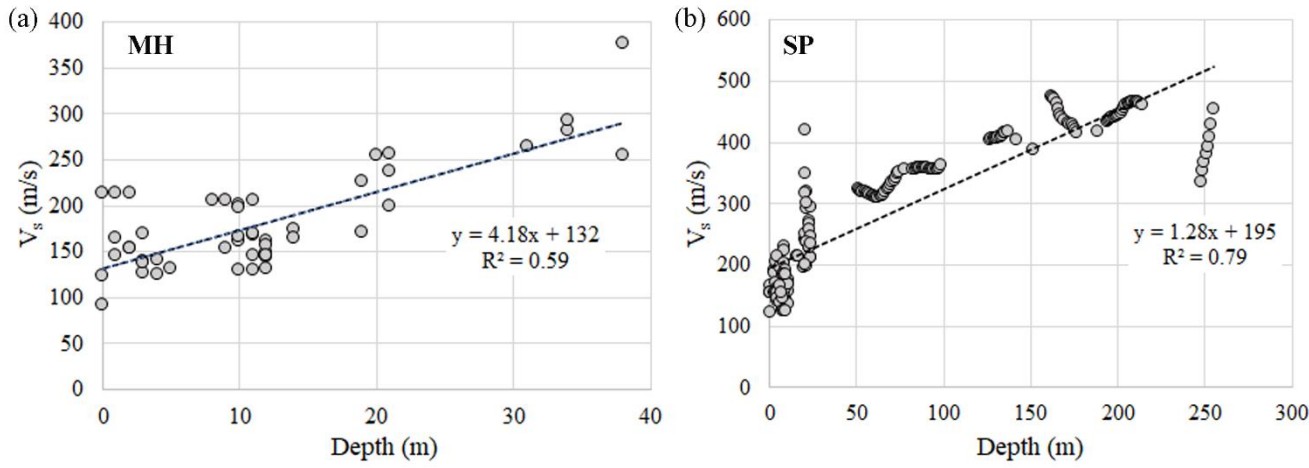

**Figure 6: Scatter plot and regression line model for lithology MH (a) and SP (b). Equations and R² are also reported.**