# Peer review of "PERL: a dataset of geotechnical, geophysical, and hydrogeological parameters for earthquake-induced hazards assessment in Terre del Reno (Emilia Romagna, Italy)"

_Natural Hazards and Earth System Sciences, 2022_

## Author Response (AR1)

Dear Referee #1,

Thank you for you accurate and constructive review. Every suggestion has been considered and implemented in the text to enhance the manuscript. With regard to the specific questions:

1) Insert the areas affected by liquefaction

Punctual and linear liquefaction effects have been reported in Figure 5.

2) few sentences on geological setting are necessary

A chapter presenting the geological setting of the study area has been integrated in the manuscript.

3) Present a geological cross-section of the study area

A geological cross-section of the study area has been integrated to Figure 1.

4) Figure 2 should be made of 2 maps, one for point investigations and another for linear, as the latter are not visible

As suggested, Figure 2 (Figure 3 of the revised version of the manuscript) has been modified to make all the investigations more visible. More in particular, linear investigations have been reported in two separated panels (Figure 3b and Figure 3c)

5) Several typos are present in the manuscript, please check

All the typos have been checked and corrected.

6) In the conclusions, provide more comments on how you used or will use these data in the future.

Conclusions have been extensively modified to provide information on existing and potential uses of PERL dataset.

Dear Referee #2,

thank you for your careful and constructive review of the manuscript. As suggested, the introduction has been modified and available literature on soil liquefaction issues after major earthquakes has been integrated in the manuscript. To spotlight the potentials and the uniqueness of this dataset, two applications have been presented in chapter 5. Moreover, conclusions have been extended to present implications and existing applications of PERL dataset.

In brief, every suggestion has been considered and implemented in the text to improve the quality of the manuscript.

---

## Author Response (AR2)

Dear Referee #1,

thank you for your careful revision of the manuscript. All your suggestions have been accepted and integrated in the text. In particular,

1) A further impovement could be to dress the map in fig. 5 with contour lines or hillshade structures and infrastructures etc.) to better locate the study area.
   Figure 5 has been dressed in counter lines and anthropic structures and infrastructures have been integrated to better locate the area.

2) In fig.6 insert r2 and equation.
   As suggested, $R^2$ and equations have been added to Figure 6.

3) Beware of some english typos, i.e. Gutam in Gautam etc.
   The manuscript has been carefully checked to correct all the typos.